# Pharmacogenomic Profiling of Cisplatin-Resistant and -Sensitive Human Osteosarcoma Cell Lines by Multimodal Targeted Next Generation Sequencing

**DOI:** 10.3390/ijms231911787

**Published:** 2022-10-04

**Authors:** Claudia Maria Hattinger, Chiara Casotti, Maria Pia Patrizio, Silvia Luppi, Leonardo Fantoni, Katia Scotlandi, Toni Ibrahim, Massimo Serra

**Affiliations:** 1Osteoncology, Bone and Soft Tissue Sarcomas and Innovative Therapies, IRCCS Istituto Ortopedico Rizzoli, 40136 Bologna, Italy; 2Department of Experimental, Diagnostic and Specialty Medicine (DIMES), University of Bologna, 40126 Bologna, Italy; 3Laboratory of Experimental Oncology, IRCCS Istituto Ortopedico Rizzoli, 40136 Bologna, Italy

**Keywords:** osteosarcoma, next generation sequencing, cisplatin resistance, single nucleotide polymorphism, pharmacogenomics

## Abstract

Cisplatin (CDDP) is a drug for high-grade osteosarcoma (HGOS) treatment. Several germline pharmacogenetic studies have revealed associations between single nucleotide polymorphisms (SNPs) and CDDP-based therapy response or CDDP-related toxicity in patients with HGOS. Whether these variants could play a biological role in HGOS cells has not been studied so far. The aim of this study was to explore 28 SNPs of 14 genes in 6 CDDP-resistant and 12 drug-sensitive human HGOS cell lines. An innovative multimodal targeted next generation sequencing (mmNGS) approach with custom primers designed for the most commonly reported SNPs of genes belonging to DNA repair, CDDP transport or detoxification, or associated with CDPP-related toxicity was applied. The mmNGS approach was validated by TaqMan genotyping assays and emerged to be an innovative, reliable tool to detect genetic polymorphisms at both the DNA and RNA level. Allele changes in three SNPs (*ERCC2* rs13181 and rs1799793, *ERCC1* rs11615) were identified on both DNA and RNA derived libraries in association with CDDP resistance. A change of the *GSTP1* rs1695 polymorphism from AA to AG genotype was observed in the RNA of all six CDDP-resistant variants. These SNPs emerged to be causally associated with CDDP resistance in HGOS cells.

## 1. Introduction

High-grade osteosarcoma (HGOS), the most common malignant tumor of bone, is treated by surgery and systemic neo-adjuvant multidrug chemotherapy [1,2]. Cisplatin (CDDP), together with high-dose methotrexate and doxorubicin, is invariably included in standard chemotherapy for this tumor [1,2].

Pharmacogenetic studies have revealed several single nucleotide polymorphisms (SNPs) of genes belonging to either DNA repair, drug transport, folate metabolism, and detoxification pathways to be associated with therapy-related parameters in HGOS, as survival and drug response, or development of drug-associated toxicity [3]. The general goal of these studies was the identification of genomic variations associated with drug response or adverse toxicities, which may provide useful information to improve treatment efficacy and simultaneously reduce the risk of chemotherapy-related toxicities [4].

In the last decade, pharmacogenomic approaches have been increasingly applied to the study of HGOS providing a series of interesting insights related to genetic polymorphisms, which may be causally related to drug resistance or susceptibility to develop treatment-related adverse toxicities [5,6]. However, all these indications must be further confirmed because the polymorphic gene status was revealed almost only in patients’ normal (germline) cells at the DNA level without providing information on how these changes were maintained at the RNA level, and influenced RNA and protein expression in tumor cells. The aim of this study was to explore the genotype status of 28 SNPs in 14 genes related to processes involved in DNA repair, CDDP transport and detoxification, or involved in CDDP-related toxicity in a panel of 6 CDDP-resistant and 12 drug-sensitive human HGOS cell lines (Table A1). In particular, we focused our study on both pharmacogenetic (germline) and pharmacogenomic (tumor-associated, somatic) markers, which had been indicated to influence treatment response and susceptibility to CDDP-related ototoxicity in HGOS patients, thus appearing as promising candidates for a translation to clinical practice. This selection was performed by taking into consideration the body of evidence reported so far, which has also been recently reviewed [3].

This analysis was performed by using an innovative multimodal targeted next generation sequencing (mmNGS) approach that allowed for the contemporary study of the selected SNPs on both DNA- and RNA-derived libraries. Data obtained by mmNGS on DNA-derived libraries were validated by TaqMan genotyping. RNA expression level of the 14 genes in CDDP-resistant variants compared to their parental cell lines was also determined. Heatmap analysis was performed, including all CDDP-resistant and drug-sensitive cell lines.

## 2. Results

### 2.1. Validation of Custom Multimodal NGS Panel

Data obtained for the 28 SNPs on DNA-derived libraries by the custom mmNGS approach were validated by TaqMan genotyping in 24/28 SNPs. Table 1 shows for each cell line the genotype status of all 28 SNPs, which were identified to be either heterozygous or homozygous by sequencing compared to the reference sequence (Table A2).

Variants with an allele frequency greater than 3% were considered reliable. Those SNPs that were homozygous wild-type for the reference allele were not reported in Table 1.

By comparing the data obtained from sequencing and genotyping, we found that for 11/18 (61%) cell lines data obtained by both techniques matched in 100% of the SNPs, whereas in 4/18 (22%) cell lines the match ranged from 90 to 93%, and was below 90% in 3/18 (17%) cell lines. The fact that 39% of the cell lines did not show a complete match could be explained by the presence of different subpopulations within the same cell line.

Interestingly, for five SNPs, *ABCC2* rs17222723, *ACYP2* rs1872328, *TPMT* rs12201199, rs1142345, and rs1800460 the homozygous wild-type genotype was identified in all 18 cell lines.

### 2.2. DNA SNP Evaluation in Relation to Level of CDDP Resistance

#### 2.2.1. Comparison between U-2OS CDDP-Resistant Variants to Parental U-2OS Cell Line

The comparison of polymorphisms identified in the group of U-2OS CDDP-resistant variants in comparison with their parental cells, identified two polymorphisms of the *ERCC2* gene, (rs13181 and rs1799793), which exhibited a genotype change in relation to the acquisition of CDDP resistance (Table 2).

In the CDDP-sensitive, parental cell line and in the two variants with the lower level of CDDP resistance (U-2OS/CDDP300 and U-2OS/CDDP1µg), the genotype of *ERCC2* rs13181 was heterozygous variant (GT), while in the variant with the highest resistance level (U-2OS/CDDP4µg) the genotype of the polymorphism shifted to homozygous wild-type (TT).

Figure 1 shows the graphical representation of the mmNGS data obtained by the DNA variant calling identifier tool of the CLC Genomics Workbench (GWB) analysis.

The data obtained for these SNPs by TaqMan genotyping and mmNGS were concordant for all cell lines except for *ERCC2* rs13181 in U-2OS/CDDP1µg variant for which mmNGS reported GT but TaqMan genotyping a TT genotype. This apparent discordance may be due to the different sensitivity of the techniques and the presence of subpopulations with TT and GT genotypes. However, these data indicate that in these cells the transition toward a TT genotype is associated with development of CDDP resistance.

For *ERCC2* rs1799793, the sensitive cell line and the two U-2OS/CDDP300 and U-2OS/CDDP1µg resistant cell lines showed a heterozygous variant genotype CT, which became homozygous (CC) in the variant with the highest resistance level (Table 2 and Figure 2).

As shown in Table 2 and Figure 3, both SNPs of *ERCC2* were non-synonymous and caused amino acid changes. The SNP rs13181 caused the substitution of Lys by Gln and the rs1799793 the substitution of Asp with Asn.

#### 2.2.2. Comparison between Saos-2 CDDP-Resistant Variants to Parental Saos-2 Cell Line

The comparison of DNA variant calling data between Saos-2 CDDP-resistant variants and their parental Saos-2 CDDP-sensitive cell line identified genotype changes of *ERCC2* rs13181 and *ERCC1* rs11615 (Table 3). These genotype changes were confirmed by both TaqMan genotyping and mmNGS.

For *ERCC2* rs13181, the genotype of the detected polymorphism was heterozygous variant GT in the sensitive and the two Saos-2 resistant variants with lower resistance levels, while in the Saos-2/CDDP6µg variant the genotype changed to homozygous variant GG (Figure 4).

The same situation occurred for *ERCC1* rs11615, which was heterozygous variant GA in the sensitive cell line and the two resistant variants with lower resistance levels, whereas homozygous variant GG in Saos-2/CDDP6µg (Figure 5).

Different to the *ERCC2* rs13181 variant, which caused an amino acid change from Lys to Gln, no amino acid changes were revealed by the CLC GWB analysis for the synonymous *ERCC1* rs11615 variant (Figure 6).

### 2.3. RNA SNP Evaluation in Relation to Level of CDDP Resistance

#### 2.3.1. Comparison between U-2OS Cell Line and U-2OS CDDP-Resistant Variants

All genotype variations identified at the DNA level and described above were also identified on the RNA level, indicating that these changes had been selected and maintained during development of CDDP resistance. Differently, the *GSTP1* rs1695 SNP changed in the RNA-derived libraries of U-2OS cell line and U-2OS/CDDP1µg variant compared to the DNA-derived libraries (Table 4). The genotype of the *GSTP1* rs1695 detected on DNA remained AG in the sensitive and in the three resistant cell lines. At the RNA level, the genotype of *GSTP1* rs1695 was homozygous wild-type AA in U-2OS and heterozygous variant AG in U-2OS/CDDP300 and U-2OS/CDDP4µg variants, while in the U-2OS/CDDPP1µg variant, a multi nucleotide variant (MNV) GAT, was detected.

Interestingly, the amino acid change Ile105Val caused by the *GSTP1* rs1695 variant allele was identified by the CLC GWB in all three CDDP-resistant U-2OS variants (Table 4).

#### 2.3.2. Comparison between Saos-2 Cell Line and Saos-2 CDDP-Resistant Variants

All genotype changes identified at the DNA level were also identified on RNA except for the *GSTP1* rs1695 SNP. As with U-2OS cell lines, the rs1695 genotype was AG at DNA level whereas homozygous AA at RNA level in the Saos-2 parental cell line (Table 5). No difference was found at DNA and RNA level for all CDDP-resistant variants (Table 5).

Accordingly, the amino acid change Ile105Val caused by the *GSTP1* rs1695 variant allele was identified by the CLC GWB in all three CDDP-resistant Saos-2 variants with the AG genotype in the RNA-derived libraries (Table 5).

### 2.4. RNA Expression Analysis

Targeted RNAseq was performed for the 14 genes related to either CDDP drug response or toxicity reported after CDDP therapy. The fold-change of transcripts per million (TPM), which estimates the fold-change in RNA expression, for each CDDP-resistant variant compared to its drug-sensitive parental cell line is graphically shown in Figure 7.

In U2OS-derived CDDP-resistant variants, six genes, *ABCC2, ABCC3, ACYP2, COMT, ERCC2*, and *XRCC3* emerged to be increased more than 2-fold compared to the parental U-2OS cell line, whereas four genes, *ATM, ATR, TP53*, and *XPA* were downregulated in CDDP-resistant variants. Considering all three CDDP-resistant variants together, the differential gene expression tool of the CLC GWB identified the downregulation of *ATM, ATR*, and *TP53* as significant with a Bonferroni corrected *p*-value < 0.05.

In Saos-2-derived CDDP-resistant variants, six genes were increased more than 2-fold: *ABCB1, ABCC2* and *XRCC3* in all three variants, whereas *ACYP2, COMT* and *ERCC2* only in Saos-2/CDDP300, the variant with the lowest resistance level. CDDP-resistant variants also presented downregulation of *ATM, ATR, GSTP1, TPMT,* and *XPA* genes. Evaluating all three CDDP-resistant variants together, a significant difference after Bonferroni correction with a *p*-value < 0.05 was identified for upregulation of *ABCB1* and downregulation of *TPMT* and *XPA*.

Similarities between CDDP-resistant and CDDP-sensitive cell lines were assessed by using the heatmap tool of the CLC GWB, including all 14 genes (Figure 8). Two main clusters were revealed. One consisted of two clusters formed by all six CDDP-resistant variants clearly separated from their two parental cell lines.

The 10 drug-sensitive cell lines formed the second main cluster, which was mostly separated from that of CDDP-resistant variants.

As also shown in Figure 8, the group of 14 genes resulted to be divided in 6 clusters, with genes belonging to the same family mostly grouped together.

## 3. Discussion

In this study, a custom mmNGS approach has been used to study 28 SNPs of 14 genes, contemporarily on the DNA and RNA level, in 6 CDDP-resistant and 12 CDDP-sensitive human HGOS cell lines. To our knowledge, this innovative approach has not been used so far for pharmacogenomic studies.

The successful validation of the DNA variant calling by TaqMan genotyping confirmed that this approach is an appropriate method to study even rare SNPs. Compared to genotyping by single TaqMan assays, the custom mmNGS approach is faster and also offers the possibility to identify additional SNPs mapping to the target region. Moreover, small targeted panels allow the pooling of higher sample numbers compared to whole-genome NGS. Another advantage of the mmNGS approach is the low amount of starting material that is required for library preparation, which facilitates the application of this method to tumor tissue samples. In addition, the simultaneous analysis of SNPs on the DNA and RNA level, as well as the possibility to estimate the level of RNA expression associated with the polymorphic gene status allow for a direct correlation between the genotype status with the biological function of each SNP.

The frequencies of variant alleles per cell line ranged from 9 (found in IOR/OS15 and MG-63) to 22 (detected in IOR/SARG), confirming the heterogeneity and high genetic instability of HGOS.

Particular genotype distributions were found for 9 SNPs. Five SNPs that had been reported in association with CDDP-related ototoxicity, *ACYP2* rs1872328 [7], *ABCC2* rs17222723 [8], *TPMT* rs12201199, rs1142345, and rs1800460 [9,10,11], were present only in the wild-type status in all cell lines. Two SNPs, *ABCC2* rs717620 and *GSTP1* rs1695, were found as homozygous wild-type or heterozygous but not as homozygous variant. These findings suggest that the variant allele of these seven SNPs could be of biological disadvantage in HGOS tumor cells. Differently, the two SNPs of *TP53*, rs1042522 and rs1642785, were identified either in a homozygous wild-type or variant but not heterozygous status.

The most relevant SNPs that emerged in this study to be associated with the development of CDDP resistance were *GSTP1* rs1695, *ERCC2* rs13181, *ERCC2* rs1799793, and *ERCC1* rs11615.

The *GSTP1* rs1695 was the only SNP for which the genotype changes found in the RNA-derived libraries differed from those revealed in DNA-derived libraries. Interestingly, in all five drug-sensitive cell lines with the heterozygous genotype in the DNA (U-2OS, Saos-2, IOR/10, IOR/14, IOR/18), the genotype status in the RNA was homozygous wild-type. The presence of the variant also in the RNA of all six CDDP-resistant cell lines, with the consequent amino acid change Ile105Val, strongly suggests that the AG genotype is associated with reduced CDDP response. These findings further support the previously demonstrated relevance of GSTP1 enhanced enzymatic activity in these CDDP-resistant HGOS cell lines [12]. The pharmacogenomic findings emerged from the present study thus indicate that the increase in GSTP1 activity observed in CDDP-resistant variants is correlated with the transition to the AG genotype of the rs1695 polymorphism and the consequent Ile105Val amino acid change.

This observation is concordant with the data reported in almost all germline studies. A significant association between AG+GG genotypes and poor histological response as well as decreased event-free and overall survival was observed in five studies [8,13,14,15,16] whereas one study reported the GG genotype to be associated with good response [17].

Interestingly, *GSTP1* rs1695 was excluded from further analyses in the study by Goricar and co-workers because the genotype frequencies of rs1695 were not in Hardy Weinberg Equilibrium [18]. Since their study was performed on paraffin embedded HGOS tumor tissue samples and not on DNA extracted from lymphocytes, as almost all pharmacogenetic analyses, their observation is concordant with our data obtained on HGOS cell lines.

Genotype changes in relation to CDDP resistance were found for the two non-synonymous SNPs *ERCC2* rs1799793 and rs13181 and the synonymous *ERCC1* rs11615 at the DNA and RNA level. All three SNPs have been reported to be associated with survival and toxicity, but the data are quite discordant [17,18,19,20,21,22,23,24,25,26,27,28,29].

However, the *ERCC2* rs1799793 GG genotype was reported in association with poor event-free survival compared to the GA +AA genotypes and *ERCC2* rs13181 AA with poor response to chemotherapy compared to AC+CC [18]. Our data obtained in CDDP-resistance cell lines confirm the relevance of these two SNPs and suggest that they could serve as biomarkers.

In one germline study performed on 130 patients with osteosarcoma treated with neoadjuvant cisplatin-based therapy in combination with doxorubicin, methotrexate, and ifosfamide the *ERCC2* rs13181 and *ERCC2* rs1799793 SNPs were associated with survival [28]. The authors suggested that the amino acid change that occurred as a result of the mutation reduced the ability of the enzyme ERCC2 to repair DNA thus resulting in greater efficacy of cisplatin. In our study this finding seems to be confirmed by the fact that the most resistant cell line returned to the wild-type genotype, restoring the repair capacity of the enzyme with the consequent increased resistance to the chemotherapeutic agent.

The functional consequences of *ERCC2* rs1799793 and rs13181 on the protein structure and stability have recently been elucidated [30,31]. Molecular dynamics simulation of the native ERCC2 protein and the variant protein with the substitution of Asp by Asn revealed that rs1799793 resulted in a destabilized, less active protein compared to the native.

In addition, the *ERCC2* rs13181 variant caused the loss of C-terminal alpha-helix and beta-sheet [30]. Although these secondary structures were lost, the overall folding was not disrupted, suggesting that this polymorphic variation has a less relevant impact on protein function.

For *ERCC1* rs11615, which changed to the homozygous wild-type genotype status in the Saos-2 CDDP-resistant variant with the highest level of resistance, two germline studies reported similar evidence being better survival associated with the TT compared to the CC genotype [24,25]. However, five studies reported the opposite evidence for overall survival [19,20,21,22,23].

Differential gene expression analysis identified dysregulations in CDDP-resistant variants compared to their parental cell lines suggesting that development of CDDP resistance influences not only genes of the NER pathway, which is known to be mainly responsible for the removal of CDDP-associated DNA adducts, but also genes belonging to other DNA repair mechanisms, such as *ATM* and *ATR*. It has been shown that cancer cells that are deficient in one DNA repair pathway can activate other functional repair pathways, which underlines the importance to study not only one of them for treatment optimization [32].

The biological consequence of the significant downregulation of *ATM* and *ATR* observed in Saos-2/CDDP-resistant variants is a relevant finding that needs to be further explored, since inhibitors against ATR have already been used in clinical trials in other cancers [33]. On the other hand, also the upregulation of *ACYP2* and *COMT*, although not significant, warrants attention because SNPs of these two genes had been described to be associated with ototoxicity after CDDP treatment [7,9,10,11].

In conclusion, the mmNGS approach emerged to be an innovative, reliable tool to detect genetic polymorphisms at both DNA and RNA level, allowing for the identification of genetic changes causally related to CDDP resistance in HGOS cells. Once further validated in tumor samples series, these SNPs could be useful to identify patients with reduced sensitivity to CDDP-based therapy and/or increased susceptibility to CDDP-related adverse toxicities.

## 4. Materials and Methods

### 4.1. Cell Lines

The study was performed on a panel of 12 drug-sensitive human HGOS cell lines: U-2OS, Saos-2, MG-63, and HOS (purchased from the American Type Culture Collection ATCC, Rockville, MD, USA) and IOR/OS9, IOR/OS10, IOR/OS14, IOR/OS15, IOR/OS18, IOR/OS20, IOR/MOS, IOR/SARG, which were established from tumor specimens at the Laboratory of Experimental Oncology of the Orthopaedic Rizzoli Institute [34].

The panel of 6 CDDP-resistant variants derived either from U-2OS (U-2OS/CDDP300, U-2OS/CDDP1μg, U-2OS/CDDP4μg) or Saos-2 (Saos-2/CDDP300, Saos-2/CDDP1μg, Saos-2/CDDP6μg) CDDP-sensitive cell lines, as previously reported [12]. Resistant variants were established by exposing parental cells to step-by-step increases of CDDP concentrations. The in vitro continuous drug exposure resulted in the establishment of variants resistant to 300 ng/mL CDDP (U-2OS/CDDP300 and Saos-2/CDDP300), 1 µg/mL (U-2OS/CDDP1µg and Saos-2/CDDP1µg), 4 µg/mL (U-2OS/CDDP4µg), or 6 µg/mL CDDP (Saos-2/CDDP6µg). Establishment of an adequate in vitro growth at each new CDDP concentration required approximately 10–12 weeks (corresponding to 8–10 in vitro passages), and variants were considered as definitely stabilized when reaching the 20th in vitro passage.

CDDP sensitivity of each cell line was expressed as IC50 (drug concentration resulting in 50% inhibition of cell growth after 96 h of in vitro treatment). The fold-increase in CDDP resistance of each variant was determined by comparing its IC50 value with that of its corresponding parental cell line and, as previously described, ranged from 4.0- to 62.5-fold for U-2OS variants and from 7.4- to 112.1-fold for Saos-2 variants [12].

All cell lines were cultured in Iscove’s modified Dulbecco’s medium (IMDM) added with 10% fetal bovine serum (Biowhittaker Europe, Cambrex-Verviers, Belgium) and maintained in a humified atmosphere with 5% CO_2_ at 37 °C. Drug resistant variants were continuously cultured in the presence of the CDDP concentrations used for their selection. Cell pellets were prepared according to standard procedures when cells were confluent, snap-frozen, and stored at −80 °C.

DNA fingerprint analyses were performed for all cell lines using 17 polymorphic short tandem repeat sequences confirming their identity.

### 4.2. Extraction of Nucleic Acids

DNA and RNA were simultaneously isolated and purified from the same pellet obtained from each cell line by using the AllPrep DNA/RNA mini kit (Qiagen, Hilden, Germany) according to the manufacturers’ instructions. During this process, the DNA and RNA were isolated from the entire sample by passing the lysate first to the AllPrep DNA spin column to isolate high molecular weight total genomic DNA and through the AllPrep RNA spin column to isolate total RNA.

A DNA and RNA quality check was performed for all samples by spectrophotometry (NP-80, Implen, Munich, Germany). All RNA samples were run on a 2100 Bioanalyzer system (Agilent, Santa Clara, CA, USA) using the RNA 6000 kit (Agilent, Santa Clara, CA, USA).

### 4.3. Custom Multi-Modal Targeted Next Generation Sequencing (mmNGS)

Library preparation was performed according to the QIAseq Multimodal Panel handbook v06/2020 (Qiagen, Hilden, Germany) for small panels. The primers for the libraries derived from DNA were designed for 28 SNPs of 14 genes related to DNA repair, CDDP transport and detoxification, and TP53 (Table A2). For the libraries prepared from RNA, primers were designed for the SNPs mapping to exons of the 14 genes, thus allowing RNA variant calling. The specific design of the RNA panel enabled also RNA expression analysis of these 14 genes (technical service Qiagen, Hilden, Germany). This approach uses integrated unique molecular indices (UMIs) which improves the specificity of variant detection.

DNA- and RNA-derived libraries were prepared for all 18 cell lines. Prior to library preparation, the nucleic acid concentrations were determined fluorometrically by Qubit high-sensitivity assays on a Qubit reader version 4.0 (Thermo Fisher Scientific by Life Technologies Italia, Monza, Italy).

For library preparation, the input amount was 40 ng of DNA and 100 ng of RNA. All libraries were run on a 2100 Bioanalyzer system (Agilent, Santa Clara, CA, USA) using the High Sensitivity DNA kit (Agilent, Santa Clara, CA, USA) to check the profile of the samples. The fragment lengths of all libraries ranged between 400 and 600 base pairs, as expected according to the protocol.

In order to provide an accurate quantification of the amplifiable libraries, the QIAseq Library Quant Assay kit (Qiagen, Hilden, Germany) was performed on a real-time PCR system (7900HT Fast Real-time PCR system; (Thermo Fisher Scientific by Life Technologies Italia, Monza, Italy) for all of them.

For sequencing, libraries were diluted to 1.2 pM, pooled together and analyzed by paired-end sequencing on a NextSeq 500 instrument (Illumina Inc., San Diego, CA, USA) using a mid-output reagent kit v2.5 (300 cycles) with a custom sequencing primer provided with the library preparation kit.

### 4.4. mmNGS Data Analysis by CLC Genomics Workbench

All bioinformatic analyses were performed using the CLC GWB software (Qiagen Bioinformatics, Aarhus, Denmark) v22.04. FastQ files were downloaded from the BaseSpace cloud (Illumina Inc., San Diego, CA, USA) and imported in the CLC GWB (Qiagen Bioinformatics, Aarhus, Denmark).

For the detection of DNA variants and gene expression, the FastQ files were analyzed using the Biomedical Genomics Analysis plugin running the Qiaseq Multimodal Analysis workflow.

The DNA and RNA reads were aligned to the human genome hg38 reference sequence and filtered using a coverage of 100× and a variant allele frequency (VAF) higher than 3%.

For RNA variant calling a custom workflow was provided by the Qiagen bioinformatics support. This workflow worked with UMIs, mapped the reads on the human genome hg38 and filtered with specific parameters for rare RNA variant calling.

For differential gene expression analysis between the groups of drug-resistant variants and their respective parental cell line, the differential expression tools of the CLC GWB were used and changes with a Bonferroni corrected *p*-value < 0.05 were considered significant.

For hierarchical clustering analysis the tool for creating heatmaps of the CLC GWB was used with Euclidean distance and complete linkage.

### 4.5. SNP Genotyping by Real-Time PCR

In total, 24 of the 28 selected polymorphisms were validated by real-time genotyping PCR (Table A2). TaqMan SNP genotyping assays (Thermo Fisher Scientific by Life Technologies Italia, Monza, Italy) or drug metabolizing enzymes (DMEs) assays, which had functionally been tested, were used to validate the performance of the mmNGS approach.

The genotyping experiments were performed according to standard protocols using 10 ng DNA as input material using the VIIA 7 DX realtime PCR system (Thermo Fisher Scientific by Life Technologies Italia, Monza, Italy) and the results were analyzed with the TaqMan Genotyper software (Thermo Fisher Scientific by Life Technologies Italia, Monza, Italy), which generated allelic discrimination cluster plots to determine the genotype of each SNP.

## Figures and Tables

**Figure 1 ijms-23-11787-f001:**
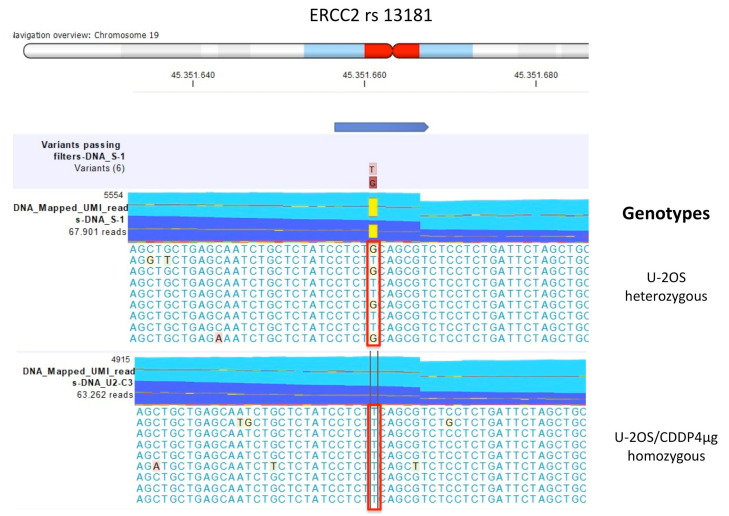
ERCC2 rs13181 genotypes identified by multimodal targeted next generation sequencing (mmNGS) in U-2OS cell line and U-2OS/CDDP4μg variant. Mismatched nucleotides were evidenced by the CLC Genomics Workbench (GWB) with a background color according to the Rasmol color scheme. The red square box indicates the polymorphism alleles.

**Figure 2 ijms-23-11787-f002:**
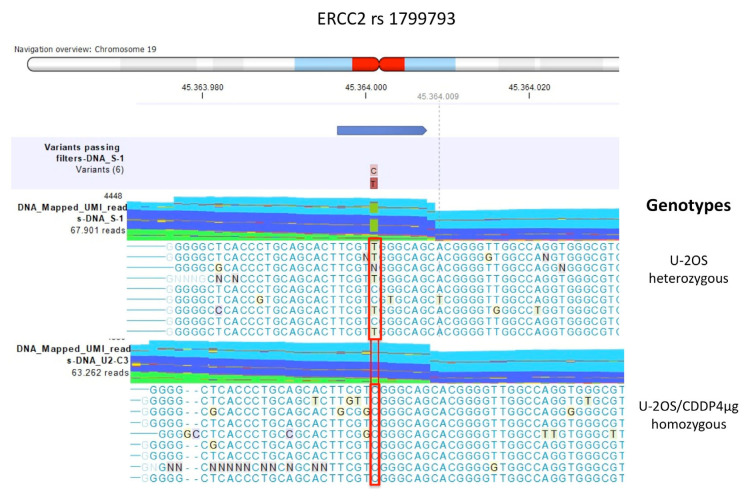
ERCC2 rs1799793 genotypes identified by multimodal targeted next generation sequencing (mmNGS) in U-2OS cell line and U-2OS/CDDP4μg variant. Mismatched nucleotides were evidenced by the CLC Genomics Workbench (GWB) with a background color according to the Rasmol color scheme. The red square box indicates the polymorphism alleles.

**Figure 3 ijms-23-11787-f003:**
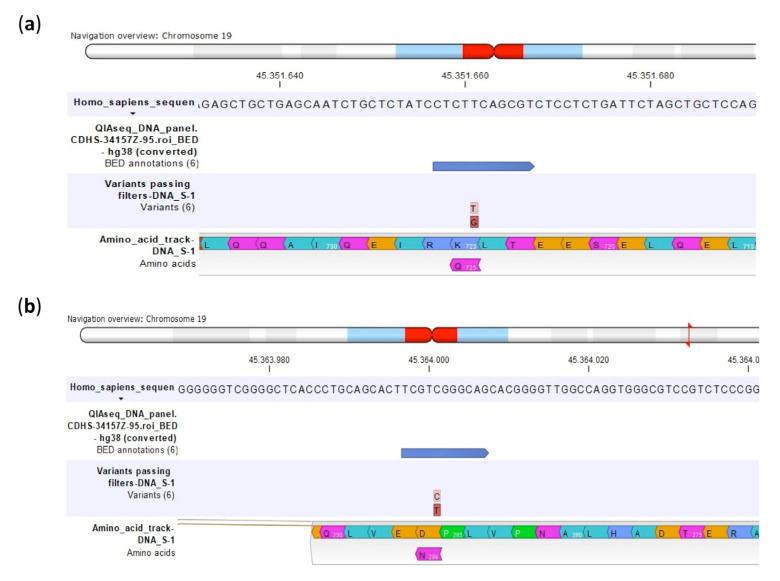
Amino acid changes identified by multimodal targeted next generation sequencing (mmNGS) in the U-2OS cell line for the ERCC2 rs13181 (**a**) and ERCC2 rs1799793 (**b**). Amino acids are colored by the CLC Genomics Workbench (GWB) according to the Rasmol color scheme.

**Figure 4 ijms-23-11787-f004:**
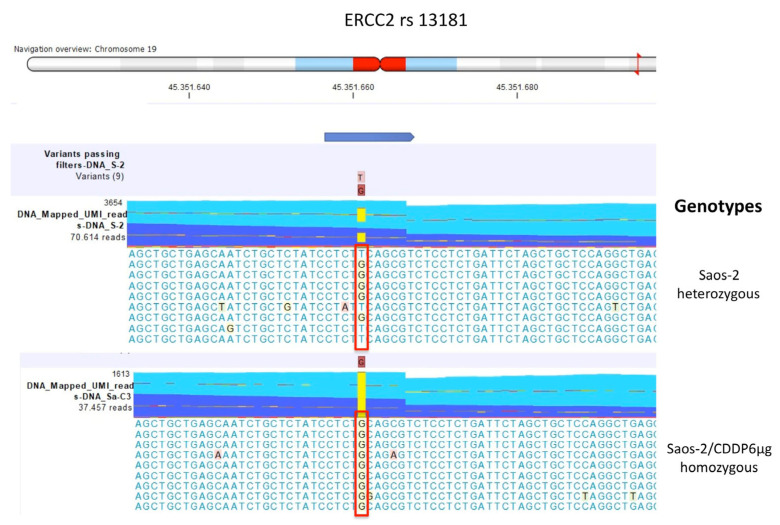
ERCC2 rs13181 genotypes identified by multimodal targeted next generation sequencing (mmNGS) in Saos-2 cell line and Saos-2/CDDP6μg variant. Mismatched nucleotides are evidenced by the CLC Genomics Workbench (GWB) with a background color according to the Rasmol color scheme. The red square box indicates the polymorphism alleles.

**Figure 5 ijms-23-11787-f005:**
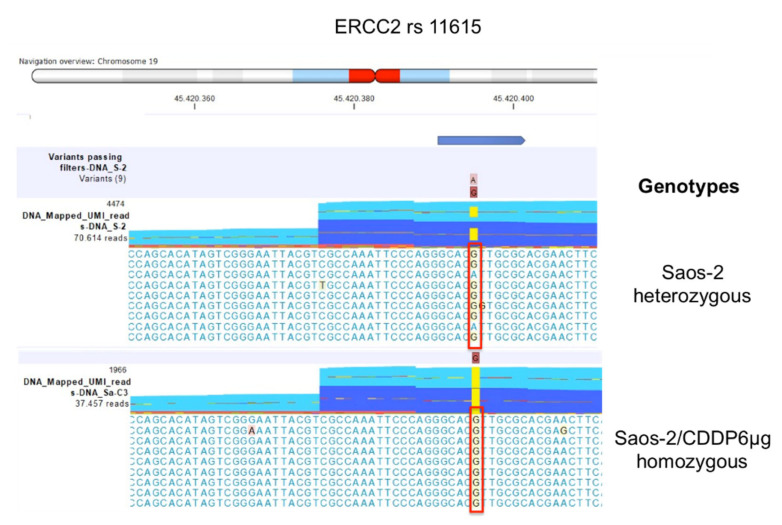
ERCC1 rs11615 genotypes identified by multimodal targeted next generation sequencing (mmNGS) in Saos-2 cell line and Saos-2/CDDP6μg variant. Mismatched nucleotides are evidenced by the CLC Genomics Workbench (GWB) with a background color according to the Rasmol color scheme. The red square box indicates the polymorphism alleles.

**Figure 6 ijms-23-11787-f006:**
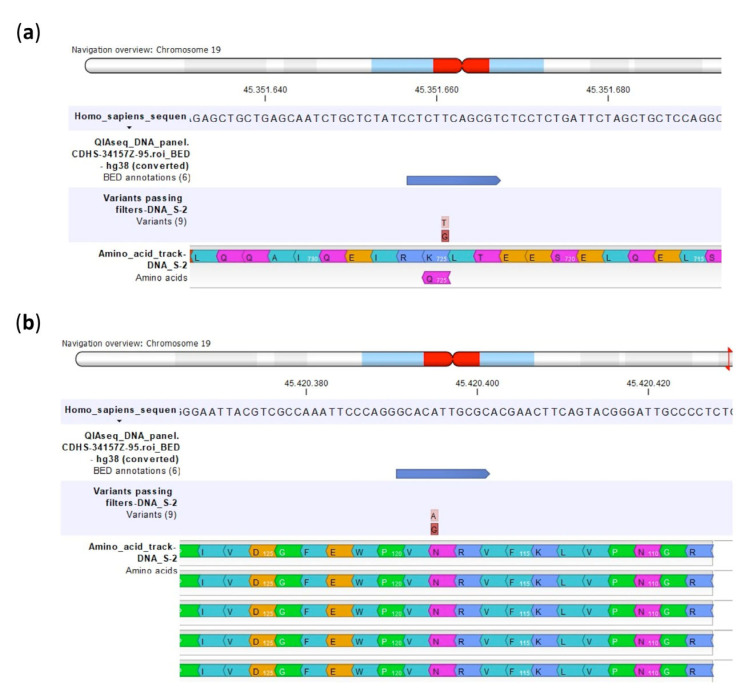
Amino acids indicated by multimodal targeted next generation sequencing (mmNGS) in the Saos-2/CDDP1μg variant for the non-synonymous ERCC2 rs13181 (**a**) and the synonymous ERCC2 rs11615 (**b**). Amino acids are colored by the CLC Genomics Workbench (GWB) according to the Rasmol color scheme.

**Figure 7 ijms-23-11787-f007:**
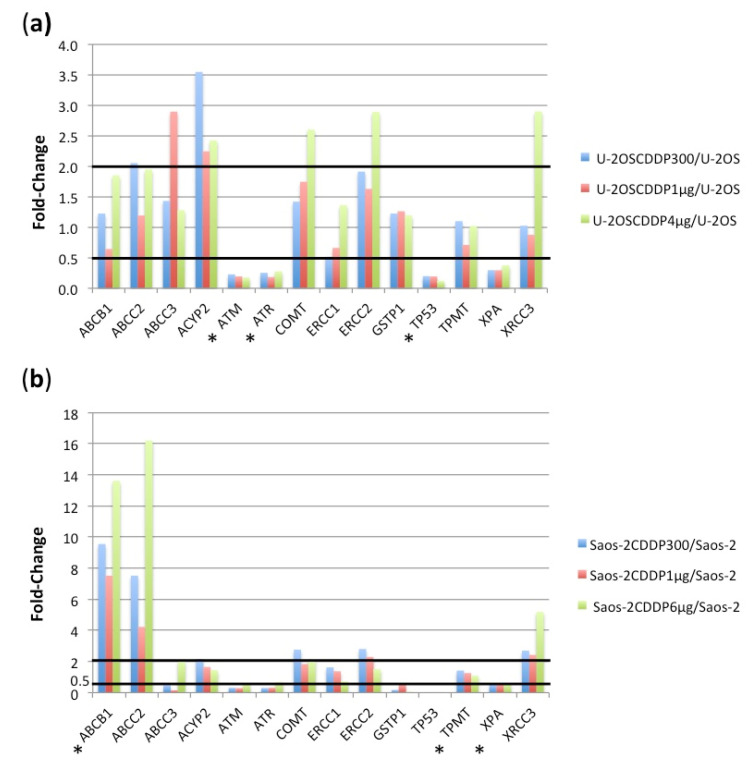
Fold-changes of RNA expression in U-2OS cisplatin (CDDP)-resistant variants (**a**) and Saos-2 CDDP-resistant variants (**b**) compared to their parental cell lines. Thick lines indicate thresholds for overexpression (2-fold) and under-expression (0.5-fold). * indicates those genes that were significantly differentially expressed in the groups of U-2OS and Saos-2 CDDP-resistant variants compared to their respective parental cell line.

**Figure 8 ijms-23-11787-f008:**
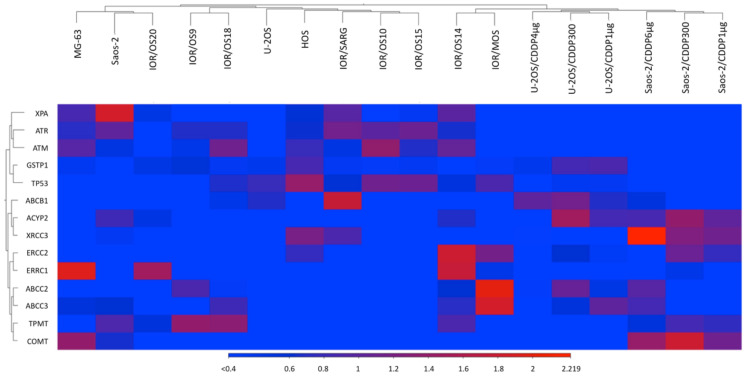
Heatmap plus hierarchical clustering analysis of 12 drug-sensitive and 6 cisplatin (CDDP)-resistant human osteosarcoma cell lines generated from RNAseq data using the CLC Genomics Workbench (GWB) tool.

**Table 1 ijms-23-11787-t001:** Variants of 28 polymorphisms identified in 18 human osteosarcoma cell lines by multimodal targeted next generation sequencing and TaqMan genotyping analysis.

Polymorphism	U-2OS		Saos-2		IOR/OS9		IOR/OS10		IOR/OS14		IOR/OS15	
	TaqM	NGS	TaqM	NGS	TaqM	NGS	TaqM	NGS	TaqM	NGS	TaqM	NGS
ABCB1 rs1045642			GA	GA	GA	GA	GA	GA	GG	GG	GG	GG
ABCB1 rs2032582	CA	CA	CA	CA	CA	CA	CA	CA	CC	CC	CC	CC
ABCB1 rs1128503	GG	GG	GA	GA	GA	GA	GA	GA	GG	GG	GG	GG
ABCC2 rs717620							CT	CT				
ABCC2 rs2273697							GA	GA	AA	AA		
ABCC2 rs3740066	CT	CT	TT	TT			TT	TT				
ABCC2 rs17222723												
ABCC3 rs4793665					/	CT	/	TT			/	TC
ABCC3 rs1051640												
ACYP2 rs1872328												
ATM rs664677	TT	TT	CT	CT			TT	TT	TT	TT	TT	TT
ATM C11orf65 rs664143	GG	GG	GA	GA			GG	GG	GG	GG	GG	GG
ATR rs2229032									TT	TT		
ATR rs2227928			GA	GA			GG	GG	GG	GG	GG/GA	GA
COMT rs4646316	TT	TT	TT	TT			CT	CT				
COMT rs9332377												
ERCC1 rs11615			GA	GA	GA	GA	GA	GA	GG	GA		
ERCC1 rs3212986					/	AC			/	AC		
ERCC2 rs13181	GT	GT	GT	GT	GT	GT	GT	GT	GT	GT	GG	GG
ERCC2 rs1799793	CT	CT	TT	TT	CT	CT			CC	TT		
GSTP1 rs1695	AG	AG	AG	AG			AG	AG	AG	AG		
TP53 rs1042522	CC	CC							CC	CC		
TP53 rs1642785									CC	CC		
TPMT rs12201199												
TPMT rs1142345												
TPMT rs1800460												
XPA rs1800975	CT	CC	CT	CT	CC	CC					CC	CC
XRCC3;KLC1 rs861539	AG	AG	AA	AA	AA	AA	GG	AG	AA	AA		
**Polymorphism**	**IOR/OS18**		**IOR/OS20**		**IOR/MOS**		**IOR/SARG**		**HOS**		**MG-63**	
	**TaqM**	**NGS**	**TaqM**	**NGS**	**TaqM**	**NGS**	**TaqM**	**NGS**	**TaqM**	**NGS**	**TaqM**	**NGS**
ABCB1 rs1045642	GA	GA			GG	GG	GG	GA	GG	GG		
ABCB1 rs2032582	CA	CA			CC	CC	CC	CC	CC	CC		
ABCB1 rs1128503	GA	GA			GG	GG	GG	GG	GG	GG		
ABCC2 rs717620					CT	CT					CT	CT
ABCC2 rs2273697			AA	AA			GA	GA				
ABCC2 rs3740066					CT	CT	CT	CT			CT	CT
ABCC2 rs17222723												
ABCC3 rs4793665	/	TC	/	TC	/	CT	/	CT	/	TT	/	CT
ABCC3 rs1051640	/	GA			/	GA						
ACYP2 rs1872328												
ATM rs664677	CT	CT	TT	TT	CT	CT	TT	TT	TT	TT	TT	TT
ATM;C11orf65 rs664143	GA	GA	GG	GG	GA	GA	GG	GG	GG	GG	GG	GG
ATR rs2229032					CT	CT	CT	CT	TT	TT		
ATR rs2227928	GG	GG	GG	GG	GG	GG	GG	GG	GG	GG		
COMT rs4646316	CT	CT										
COMT rs9332377					/	TT						
ERCC1 rs11615			GG	GG			GG	GG	GG	GG	GA	GA
ERCC1 rs3212986							/	AA	/	AC	/	AC
ERCC2 rs13181					GT	GT					GG	GG
ERCC2 rs1799793			TT	TT			CC	TT			TT	TT
GSTP1 rs1695	AG	AG										
TP53 rs1042522	CC	CC	CC	CC	CC	CC	CC	CC	CC	CC		
TP53 rs1642785												
TPMT rs12201199												
TPMT rs1142345												
TPMT rs1800460												
XPA rs1800975	CC	CC	CC	CC			CT	CC				
XRCC3;KLC1 rs861539			AA	AA	AG	AG	AG	AG				
**Polymorphism**	**U-2OS/** **CDDP300**		**U-2OS/** **CDDP1μg**		**U-2OS/** **CDDP4μg**		**Saos-2/** **CDDP300**		**Saos-2/** **CDDP1μg**		**Saos-2/** **CDDP6μg**	
	**TaqM**	**NGS**	**TaqM**	**NGS**	**TaqM**	**NGS**	**TaqM**	**NGS**	**TaqM**	**NGS**	**TaqM**	**NGS**
ABCB1 rs1045642							GA	GA	GA	GA	GA	GA
ABCB1 rs2032582	CA	CA	CA	CA	CA	CA	CA	CA	CA	CA	CA	CA
ABCB1 rs1128503	GG	GG	GG	GG	GG	GG	GA	GA	GA	GA	GA	GA
ABCC2 rs717620												
ABCC2 rs2273697												
ABCC2 rs3740066	CT	CT	CT	CT	CT	CT	TT	TT	TT	TT	TT	TT
ABCC2 rs17222723												
ABCC3 rs4793665			/	TT	/	TT						
ABCC3 rs1051640												
ACYP2 rs1872328												
ATM rs664677	TT	TT	TT	TT	TT	TT	CT	CT	CT	CT	CT	CT
ATM;C11orf65 rs664143	GG	GG	GG	GG	GG	GG	GA	GA	GA	GA	GA	GA
ATR rs2229032												
ATR rs2227928							GA	GA	GA	GA	GA	GA
COMT rs4646316	TT	TT	TT	TT	TT	TT	TT	TT	TT	TT	TT	TT
COMT rs9332377												
ERCC1 rs11615							GA	GA	GA	GA	GG	GG
ERCC1 rs3212986												
ERCC2 rs13181	GT	GT	TT	GT			GT	GT	GT	GT	GG	GG
ERCC2 rs1799793	CT	CT	CT	CT			TT	TT	TT	TT	TT	TT
GSTP1 rs1695	AG	AG	AG	AG	AG	AG	AG	AG	AG	AG	AG	AG
TP53 rs1042522	CC	CC	CC	CC	CC	CC						
TP53 rs1642785												
TPMT rs12201199												
TPMT rs1142345												
TPMT rs1800460												
XPA rs1800975	CC	CC	CC	CC	CC	CC	CT	CT	CT	CT	CT	CT
XRCC3;KLC1 rs861539	AG	AG	AG	AG	GG	AG	AA	AA	AA	AA	AA	AA

TaqM: TaqMan; NGS: Next generation sequencing;/SNPs not validated by TaqMan genotyping.

**Table 2 ijms-23-11787-t002:** DNA single nucleotide polymorphisms with genotype and amino acid changes in U-2OS cisplatin (CDDP)-resistant variants compared to their parental cell line. Where not differently indicated, genotypes were identified by both multimodal targeted next generation sequencing (mmNGS) and TaqMan genotyping.

Cell Line	Identified Variants	
	ERCC2 rs13181	ERCC2 rs1799793
U-2OS	GT	CT
U-2OS/CDDP300	GT	CT
U-2OS/CDDP1μg	GT ^(a)^/TT ^(b)^	CT
U-2OS/CDDP4μg	TT	CC
	**Amino acid change**	
U-2OS	Lys725Gln	Asp286Asn
U-2OS/CDDP300	Lys725Gln	Asp286Asn
U-2OS/CDDP1μg	Lys725Gln	Asp286Asn
U-2OS/CDDP4μg	/	/

^(a)^ detected by mmNGS; ^(b)^ detected by TaqMan genotyping.

**Table 3 ijms-23-11787-t003:** DNA single nucleotide polymorphisms with genotype and amino acid changes in Saos-2 cisplatin (CDDP)-resistant variants compared to their parental cell line, which were identified by both multimodal targeted next generation sequencing (mmNGS) and TaqMan genotyping.

Cell Line	Identified Variants	
	ERCC2 rs13181	ERCC1 rs11615
Saos-2	GT	GA
Saos-2/CDDP300	GT	GA
Saos-2/CDDP1μg	GT	GA
Saos-2/CDDP6μg	GG	GG
	**Amino Acid Change**	
Saos-2	Lys725Gln	Synonymous
Saos-2/CDDP300	Lys725Gln	Synonymous
Saos-2/CDDP1μg	Lys725Gln	Synonymous
Saos-2/CDDP6μg	Lys725Gln	Synonymous

**Table 4 ijms-23-11787-t004:** RNA single nucleotide polymorphism (SNP) of *GSTP1* gene detected in U-2OS cisplatin (CDDP)-resistant variants compared to their parental cell line.

Cell Line	Identified Variants	
	GSTP1 rs1695	
	DNA	RNA
U-2OS	AG	AA
U-2OS/CDDP300	AG	AG
U-2OS/CDDP1μg	AG	GAT
U-2OS/CDDP4μg	AG	AG
	**Amino acid change**	
U-2OS	/	
U-2OS/CDDP300	Ile105Val	
U-2OS/CDDP1μg	Ile105Val	
U-2OS/CDDP4μg	Ile105Val	

**Table 5 ijms-23-11787-t005:** RNA single nucleotide polymorphism (SNP) of *GSTP1* gene detected in Saos-2 cisplatin (CDDP)-resistant variants compared to their parental cell line.

Cell Line	Identified Variants	
	GSTP1 rs1695	
	DNA	RNA
Saos-2	AG	AA
Saos-2/CDDP300	AG	AG
Saos-2/CDDP1μg	AG	AG
Saos-2/CDDP6μg	AG	AG
	**Amino acid change**	
Saos-2	/	
Saos-2/CDDP300	Ile105Val	
Saos-2/CDDP1μg	Ile105Val	
Saos-2/CDDP6μg	Ile105Val	

## Data Availability

The data generated and analyzed in this study contained within the article or supplementary material will be made available by the corresponding author to qualified researchers, upon justified request.

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
