# Peer review of "Pharmacogenomic Profiling of Cisplatin-Resistant and -Sensitive Human Osteosarcoma Cell Lines by Multimodal Targeted Next Generation Sequencing"

_ijms, 2022, doi:10.3390/ijms231911787_

Round 1

Reviewer 1 Report

1. This is a well-designed study to figure out the SNPs in the HGOS cell lines for genetic varients. However, the authors may provides more information regarding the genes selection for readers' understanding.

2. This is a pharmacogenetics study for the cisplatin resistant cells. Since the NGS techniques are the powerful method for pharmacogenomics study, the major discussion may focus on the interactions among all genes and chromosome regulations. If the study and discussion target to the interactions among all genes, the results might benefit to the clinical physicians.

Author Response

Reply to Reviewer 1

1) This is a well-designed study to figure out the SNPs in the HGOS cell lines for genetic varients. However, the authors may provides more information regarding the genes selection for readers' understanding.

AUTHORS' ANSWER: We would like to thank the Reviewer for this general positive comment about our study. By taking in consideration this remark, we have implemented the description of the gene selection process in the Introduction section.

2) This is a pharmacogenetics study for the cisplatin resistant cells. Since the NGS techniques are the powerful method for pharmacogenomics study, the major discussion may focus on the interactions among all genes and chromosome regulations. If the study and discussion target to the interactions among all genes, the results might benefit to the clinical physicians.

AUTHORS' ANSWER: We agree that NGS techniques are a powerful method for pharmacogenomic studies because they allow the application of several tools for studying also gene interactions, as suggested by the Reviewer. For our study a set of polymorphisms and genes has been selected for which data had been reported regarding their possible role in CDDP-related resistance or association with collateral toxicity in osteosarcoma patients. Therefore, we did not perform further computational analysis.

However, to accomplish this Reviewer’s remark, which was surely useful to improve the quality of our work, we have implemented the Discussion with information regarding the possible interactions of genes belonging to different DNA repair pathways and have also cited two new references Kiss IJMS 2022 (ref. 32) and Rocha Clinics 2018 (ref. 33). In addition, the paragraph regarding the genotype distributions was rewritten.

Once validated our data in tumor samples and correlated with clinical parameters, we hopefully will be able to provide more conclusive insights which can also be beneficial in the clinical setting.

Reviewer 2 Report

In this study Hattinger et al. examined the SNPs of 14 genes in CDDP-resistant and -sensitive human osteosarcoma cell lines using NGS and TaqMan genotyping assays. The authors identified the several characteristic genomic changes in the CDDP-resistant cell.

Therapeutic resistance of osteosarcoma is still one of the most critical problems in clinical settings. Therefore, the authors’ findings regarding the characteristic genetic variation of molecules possibly involved in drug resistance of osteosarcoma cells is important.

However, there are several concerns, and the authors are requested to address the following comments.

Major

1. Whether the variation of SNPs and the substitution of amino acids shown in this study could play a biological role in osteosarcoma cells is not verified in this study. The biological meanings such as decreased function of the protein at least in ERCC2 and GSTP1 should be experimentally clarified.

2. The concise summary of the process of the establishment and the characteristics of the CDDP-resistant cells used in this study, such as ‘U-2OS/CDDP4ug’, should be provided.

3. Fig. 7: Significant differences between groups should be noted.

Minor

There are several typographical and grammatical errors in the manuscript in lines 294 and 296 ‘ototoxicity’ etc.

Author Response

Reply  to Reviewer 2

1) Whether the variation of SNPs and the substitution of amino acids shown in this study could play a biological role in osteosarcoma cells is not verified in this study. The biological meanings such as decreased function of the protein at least in ERCC2 and GSTP1 should be experimentally clarified.

AUTHORS' ANSWER: We thank for the reviewer’s comment and have integrated the knowledge regarding the impact of ERCC2 polymorphisms on protein stability, which was reported in two recent studies by molecular dynamic simulations. Two new references have been added in the revised version of the manuscript, Pasqui IJMS 2022 (ref. 30) and Peissert Nat Commun 2020 (ref. 31), and the Discussion section has been implemented. Regarding the functional consequence of GSTP1 rs1695, we have discussed the data of the present study in the context of the findings of our previous work by Pasello Cancer Res 2008 (ref. 13) in the Discussion section.

2) The concise summary of the process of the establishment and the characteristics of the CDDP-resistant cells used in this study, such as ‘U-2OS/CDDP4ug’, should be provided.

AUTHORS' ANSWER: By following the Reviewer's suggestion, additional details concerning the establishment and characteristics of CDDP-resistant variants have been provided in the revised version of the Materials and Methods 4.1 section.

3) Fig. 7: Significant differences between groups should be noted.

AUTHORS' ANSWER: We thank the reviewer for this comment. To improve the clearness of Figure 7, we have added two thick lines indicating the 2-fold and 0.5-fold thresholds and highlighted with a star those genes which were identified as significantly differentially expressed in all three resistant variants compared to the parental drug-sensitive cell line. Significance determination was performed using the differential expression tools of the CLC GWB software.

Accordingly, the figure legend and the text of the Results section 2.4, the Materials and Methods section 4.4, and the Discussion section have been modified and implemented.

4) There are several typographical and grammatical errors in the manuscript in lines 294 and 296 ‘ototoxicity’ etc.

AUTHORS' ANSWER: The whole manuscript has been checked for typographical and grammatical errors. We do not understand the example of 'ototoxicity' by the Reviewer, because this term was correctly written along the whole manuscript. Instead, we found in line 241 of the original manuscript that the term CDDP-related toxicity was used instead of CDDP-related ototoxicity. This error has been fixed in the revised manuscript.

We have corrected the following typing errors: U2OS-derived to U-2OS-derived, wild type to wild-type, we have removed the “-“ in "CDDP resistance" and "next-generation".

Round 2

Reviewer 2 Report

Since the authors have not address this reviewer’s concerns experimentally, the molecular aspects of SNPs as well as the statistical significance of the data are still unclear. However, the plot of this study has been improved and is reasonable.